# Freshwater Mussel Viromes Increase Rapidly in Diversity and Abundance When Hosts Are Released from Captivity into the Wild

**DOI:** 10.3390/ani14172531

**Published:** 2024-08-30

**Authors:** Jordan C. Richard, Tim W. Lane, Rose E. Agbalog, Sarah L. Colletti, Tiffany C. Leach, Christopher D. Dunn, Nathan Bollig, Addison R. Plate, Joseph T. Munoz, Eric M. Leis, Susan Knowles, Isaac F. Standish, Diane L. Waller, Tony L. Goldberg

**Affiliations:** 1Department of Pathobiological Sciences, University of Wisconsin-Madison, Madison, WI 53711, USA; cddunn2@wisc.edu (C.D.D.); aplate@wisc.edu (A.R.P.); jtmunoz@wisc.edu (J.T.M.); 2Southwestern Virginia Field Office, U.S. Fish and Wildlife Service, Abingdon, VA 24210, USA; rose_agbalog@fws.gov; 3Aquatic Wildlife Conservation Center, Virginia Department of Wildlife Resources, Marion, VA 24354, USA; tim.lane@dwr.virginia.gov (T.W.L.); sarah.colletti@dwr.virginia.gov (S.L.C.); tiffany.leach@dwr.virginia.gov (T.C.L.); 4U.S. Geological Survey, National Wildlife Health Center, Madison, WI 53711, USA; nbollig@wisc.edu (N.B.); sknowles@usgs.gov (S.K.); 5Department of Biostatistics and Medical Informatics, University of Wisconsin-Madison, Madison, WI 53711, USA; 6La Crosse Fish Health Center, Midwest Fisheries Center, U.S. Fish and Wildlife Service, Onalaska, WI 54650, USA; eric_leis@fws.gov (E.M.L.); sirisaac_standish@fws.gov (I.F.S.); 7U.S. Geological Survey, Upper Midwest Environmental Sciences Center, La Crosse, WI 54603, USA; dwaller@usgs.gov

**Keywords:** virome, freshwater mussel, unionid, viruses, mass mortality event, die-off, aquaculture, conservation, restoration, endangered species

## Abstract

**Simple Summary:**

Freshwater mussels create habitat, filter water, and enhance food webs, but they are also among the world’s most imperiled taxa. Conservation efforts largely rely on captive propagation in which mussels are grown in protected aquaculture environments (hatcheries) for later release. Recent evidence has highlighted the importance of pathogens in population losses of freshwater mussels. In response to ongoing mass mortality events of freshwater mussels in the Upper Tennessee River Basin in Virginia and Tennessee, USA, we conducted a multi-year study to document viruses across multiple restoration sites and compare them to viruses in mussels from the hatchery. Viral communities changed greatly after mussels were released. Of the 681 viruses of the 27 families we documented, only 20 viruses were found exclusively in hatchery mussels, compared to 451 viruses found only in mussels stocked to the wild. After release, mussels rapidly acquired new viruses, and the number of viruses increased steadily over time. These findings have implications for how mussel introduction programs might be managed for greater success, for example, by incorporating acclimatization periods prior to full release.

**Abstract:**

Freshwater mussels (order: Unionida) are highly imperiled globally and are increasingly the focus of captive propagation efforts to protect and restore wild populations. The Upper Tennessee River Basin (UTRB) in Virginia is a freshwater biodiversity hotspot hosting at least 45 of North America’s ~300 species of freshwater mussels, including 21 threatened and endangered species listed under the U.S. Endangered Species Act. Recent studies have documented that viruses and other microbes have contributed to freshwater mussel population declines in the UTRB. We conducted a multi-year longitudinal study of captive-reared hatchery mussels released to restoration sites throughout the UTRB to evaluate their viromes and compare them to captive hatchery environments. We documented 681 viruses from 27 families. The hatchery mussels had significantly less viruses than those deployed to wild sites, with only 20 viruses unique to the hatchery mussels. After the hatchery mussels were released into the wild, their number of viruses initially spiked and then increased steadily over time, with 451 viruses in total unique to the mussels in the wild. We found Clinch densovirus 1 (CDNV-1), a virus previously associated with mass mortality events in the Clinch River, in all samples, but the wild site mussels consistently had significantly higher CDNV-1 levels than those held in the hatchery. Our data document substantial differences between the viruses in the mussels in the hatchery and wild environments and rapid virome shifts after the mussels are released to the wild sites. These findings indicate that mussel release programs might benefit from acclimatization periods or other measures to mitigate the potential negative effects of rapid exposure to infectious agents found in natural environments.

## 1. Introduction

Freshwater mussels (Unionida) are essential ecosystem engineers in freshwater habitats, where they influence ecosystem function via nutrient cycling [1], habitat modification [2], and food web enhancement [3]. Myriad stressors have led to global declines in freshwater mussel populations, including invasive species [4], overharvest [5], habitat destruction [6], the anthropogenic degradation of water quality [7], and mass mortality events (MMEs) of unknown etiology [8]. Such stressors have degraded populations to the extent that freshwater mussels are now among the most imperiled faunal groups globally [9]. Approximately 10% of North America’s 300 recognized species have been declared extinct in the past century, and two-thirds of those remaining are considered threatened, endangered, or vulnerable [10]. The U.S. Fish and Wildlife Service currently lists 69 species as Endangered and 20 species as Threatened under the U.S. Endangered Species Act (ESA) [11]. An additional 14 species are currently (as of 2024) proposed for listing under the ESA (11 Endangered, 3 Threatened) with many other species currently under evaluation to determine if ESA protection is warranted.

The conservation and restoration of unionids largely relies on captive propagation and/or translocation, in which large numbers of animals are raised in captivity at hatcheries or collected from healthy populations and moved to release at sites in need of population restoration [12]. Viral disease has affected hatchery propagation and aquaculture efforts worldwide, with notable examples including ostreid herpesvirus 1 (OsHV-1) and its variants in oysters [13]; infectious hypodermal and hematopoietic necrosis virus (IHHNV), white spot syndrome virus (WSSV), and penaeid shrimp myonecrosis virus (PsIMNV) in shrimp [14]; and viral hemorrhagic septicemia virus (VHSV) in finfish [15]. Aquaculture and hatchery operations are typically supported by regular animal health assessments, which include lists of known and notifiable diseases and protocols to minimize the risk of spreading pathogens between hatchery and wild populations [16,17]. Whereas health assessments and biosecurity measures are cornerstones of other aquaculture programs, they are lacking for freshwater mussel conservation efforts, largely because the state of knowledge regarding mussel pathogens has been unable to keep pace with the increasing need for propagation as a means to avoid imminent species extinctions [12]. Further compounding these issues, freshwater mussels are almost exclusively cultured as part of conservation efforts, meaning that there are no underlying economic incentives motivating disease diagnosis and prevention, as commonly occur in sport fish and food aquaculture operations [18]. Efforts to begin understanding the role of pathogens and microbes in freshwater mussel health, disease, and restoration have only begun in earnest in recent decades [19]. These studies were largely spurred by MMEs of unknown etiology affecting populations throughout the United States and Europe [20,21]. Initial efforts on this front have documented links between freshwater mussel MMEs and viruses [22,23], bacteria [24,25,26], and parasites [27,28], most of which were previously unknown.

The Upper Tennessee River Basin (UTRB) in Virginia and Tennessee is an aquatic biodiversity hotspot comprising the Powell, Clinch, and Holston Rivers. The Clinch River has experienced an annually recurring mussel MME affecting populations of the pheasantshell (*Actinonaias pectorosa*) and other species since 2016 [22]. Several previous studies have described epidemiologic surveys assessing potential pathogenic causes of the recurring Clinch River MME. Comparisons of apparently healthy and moribund pheasantshell collected from multiple sites in October–November 2017 and August–October 2018 revealed 17 previously undescribed viruses, of which 5 were associated with moribund mussels based on either higher prevalence or viral load. Only Clinch densovirus 1 (CDNV-1) had significantly higher prevalence and viral load in moribund mussels [22]. Studies of the same sample set also found that moribund mussels were consistently associated with higher bacterial loads and higher prevalence of the bacteria *Yokenella regensburgei* and *Aeromonas* spp. [24,25]. Metabolomic analysis revealed differences between apparently healthy and moribund mussels, with the latter group associated with catabolic processes [29]. Follow-up studies in 2019–2020 further documented the relationship between *Y. regensburgei*, *Aeromonas* spp. and mussel morality [30], and another study characterized pathology associated with yokenellosis [31]. 

The goals of this study were to (1) assess whether the exposure of presumably healthy hatchery-raised juvenile mussels to the conditions within and around the die-off zone in the Clinch River induced morbidity/mortality, (2) assess how mussel viromes shift when hatchery-raised juveniles are transferred to wild restoration sites, (3) evaluate the distribution of CDNV-1 in hatchery and wild sites in the UTRB, and (4) assess spatial and temporal effects on mussel viromes released at wild restoration sites in the UTRB. We conducted a multi-year, longitudinal study using captive-raised juvenile pheasantshell mussels placed in sites throughout the UTRB as sentinel animals. This study design allowed us to characterize the distribution of mussel viruses in the environment and the dynamics of virus acquisition and loss for mussels released to wild sites for population restoration.

## 2. Materials and Methods

### 2.1. Experimental Design and Sampling

Pheasantshell mussels were propagated at the Virginia Department of Wildlife Resources’ Aquatic Wildlife Conservation Center (AWCC) in Marion, Virginia. Gravid female broodstock mussels were collected from the Clinch River for propagation as part of an ongoing restoration project. Glochidia were collected from adult females and placed in small aerated tanks with host fish to allow infestation on fish gills. Fish were then removed from infestation tanks and held in 100 L round, recirculating tanks until juvenile mussels excysted following metamorphosis. Newly transformed juveniles were initially held in indoor rearing systems with water sourced from a pond filled from the nearby South Fork Holston River. Indoor units used flow-through pond water filtered to 5 µm and supplemented with a commercial shellfish diet consisting of Reed Mariculture shellfish diet 1800^®^ and nano 3600™ (Reed Mariculture, Campbell, CA, USA). After initial culture and growth to ~3–5 mm length, mussels were transferred to an outdoor recirculating system with water fed from a second pond. The pond was a static system maintaining controlled algal growth via fertilization to provide food for mussels, and water was periodically added from the South Fork Holston River to maintain a consistent water level and maintain suitable temperatures (i.e., <30 °C) for cultured mussels. 

In 2020, we conducted an 8-week study of mussels deployed to an upstream and downstream site in the Clinch River, CP and SYC, respectively (Figure 1). The study period was based on the previously observed high mortality rates from a recurring mussel die-off in the preceding 4 years affecting SYC but not CP. We used 56 hatchery animals from a single rearing cohort, all of which were individually tagged for identification and randomly assigned to study sites and sampling dates. On 9 September 2020, we deployed 20 mussels to each of the 2 study sites. Mussels were held in concrete silos, which allow river water to flow through a mesh screen while preventing escape (Appendix A). Each site had 4 silos containing 5 mussels per silo. Eight mussels were collected from the hatchery on the first day of the study as a baseline sample. We retrieved 5 mussels from each site (1 mussel from each silo, plus 1 additional mussel from 1 silo) at approximately 2-week intervals until the last individuals were collected on 11 June 2020 (day 59). On the fourth and final sampling event (6 November 2020), we collected the last 5 mussels from each silo and sampled 8 additional mussels held at the hatchery for comparison. We measured shell length for each mussel placed in silos at the outset of the study and upon retrieval. 

In 2021, we conducted a separate experiment with increased sampling interval length, number of study sites, and duration of deployment. We selected 5 study sites from the UTRB in Virginia. In the Clinch River, we again placed silos at the SYC study site, as well as a site much farther upstream (BEN). The Powell River (POW) watershed is adjacent to the Clinch River to the west, and we placed silos near the Hwy 833 bridge crossing. The Holston River watershed is adjacent to the Clinch River watershed to the east. We placed silos in the Middle Fork Holston River (MFH) and in the North Fork Holston River (NFH). We selected 116 mussels from a hatchery cohort and randomly assigned them to the 5 river sites or the control group to remain at the hatchery. From 21–23 June 2021, we placed 2 silos with 10 mussels each at the 5 study sites and sampled 8 mussels from the cohort remaining at the hatchery as a baseline sample. We planned to retrieve a random subsample of 5 mussels from each of the study sites at 1-month intervals for 4 months and the final 8 hatchery mussels upon study completion. To account for animal losses and maintain the full temporal duration of the experiment, we skipped the planned third sample collection interval, resulting in sample collections after 1, 2, and 4 months of deployment to study sites. Throughout the 2021 study, we also collected 5 mussels from the hatchery control cohort contemporaneously with each study site collection event. Using calipers, we measured shell length along the longest axis for mussels placed in silos at the outset of the study and upon retrieval. All mussels were individually tagged for identification and randomly assigned to study sites and sampling dates.

For each mussel removed from a silo for study, we removed soft tissues from shells immediately using sterile instruments and placed them in separate microcentrifuge vials in RNAlater (Invitrogen, Waltham, MA, USA) in a 10:1 ratio of RNAlater:tissue. Samples were held at ~4 °C for 24 h to allow the preservative to fully penetrate tissue, and then they were transferred to −20 °C for the remainder of the study period. After the final samples were collected, all samples were transferred to the University of Wisconsin–Madison and held at −80 °C until extraction and sequencing. Including the hatchery cohort samples, we collected tissues from 27 unique combinations of sampling site and location (hereafter “sampling event”) across the two study years. 

### 2.2. Viral Nucleic Acid Extraction and Sequencing

We pooled mussels from each sample replicate (i.e., mussels collected from a specific site and time point). In most cases, we selected all 5 mussels available from each sampling event for sequencing. For sampling events with >5 tissues available, we randomly selected 5 individuals from among those available. Three sample replicates were sequenced using <5 mussels due to animals lost during flooding events (refer to Appendix A for complete details of samples available and selected for each sample pool). We used a sterile 3 mm disposable biopsy punch (Robbins Instruments, Houston, TX, USA) to collect a tissue sample from the visceral mass of each animal selected for sequencing. 

We processed mussel tissues for virus characterization using previously described methods [22,32]. Briefly, we homogenized pooled tissues using Qiagen PowerBead Tubes (Qiagen, Hilden, Germany) with 2.8 mm metal beads in 1.0 mL of Hank’s balanced salt solution. We then centrifuged 300 µL of homogenized tissue at 10,000× *g* for 10 min to pellet debris, transferred the supernatant to new, sterile vials, and concentrated virus particles by centrifugation at 25,000× *g* for 3 h. We extracted total nucleic acids using the QIAamp MinElute Virus Spin Kit (Qiagen, Hilden, Germany). We used the Superscript IV system (Thermo Fisher, Waltham, MA, USA) with random hexamers and the NEBNext Ultra II Non-Direction RNA Second Strand Synthesis Module to convert RNA to double-stranded cDNA and prepared DNA libraries using the Nextera XT DNA Library Preparation Kit (Illumina, San Diego, CA, USA). We sequenced libraries on a MiSeq instrument (V3 chemistry, 600 cycle kit; Illumina, San Diego, CA, USA). 

### 2.3. Sequence Analysis and Phylogenetics

We quality-trimmed demultiplexed reads to ≥Q30 using CLC Genomics Workbench version 20.1 (Qiagen, Hilden, Germany) and discarded reads shorter than 50 nucleotides (nt). We filtered remaining reads to remove laboratory contaminants, low-complexity regions, and eukaryotic reads using an in-house database, which includes known contaminants from sequencing blanks, ribosomal RNA sequences, and the NCBI UniVec database. We used metaSPAdes v3.15.2 [33] to assemble reads into contiguous sequences (contigs) for each sequencing pool. We collated all contigs > 500 nt from all samples and used CD-HIT-EST [34] to cluster redundant sequences using a threshold of ≥97% nt identity. We kept the longest representative of each cluster and trimmed the resulting list to contigs ≥ 1000 nt for subsequent processing. We evaluated potential viral contigs by running BLASTx queries (6-frame translation) against a custom in-house database of representative viral proteins. We removed contigs with no significant similarity to viral sequences and sequences matching prokaryotic viruses (phage). We then used DIAMOND [35] to query remaining contigs using a BLASTx search against the GenBank non-redundant protein (nr) database and nBLAST [36] to query contigs using the nt database. 

After identifying viral sequences in the dataset, we used Cenote-taker2 in annotation mode to identify and annotate open reading frames (ORFs) and assign taxonomy of the closest relative via BLASTx search. We manually checked ORFs and corrected instances of misidentified ORFs. We then annotated sequences further using the Conserved Domain Database (CDD, [37]), Cenote-Taker 2 (in annotation mode) [38], and the RPS-BLAST + rpsbproc tools [39]. We checked for direct terminal repeats on putative circular viruses and circularized genomes when appropriate.

### 2.4. Statistical Analysis

After identification of viral contigs, we mapped quality-trimmed sequence reads to all contigs using CLC’s map reads function with a stringency of 90% sequence identity over 90% read length. We calculated viral reads per million per kilobase of target sequence (vRPM/kb), a measure of viral concentration normalized by contig length to account for differing target sequence lengths for each virus [40]. For each sequencing pool, we also calculated a similar measure of cumulative viral concentration (hereafter “viral intensity”) by summing all reads mapped to all virus contigs and normalizing by the cumulative length of all summed target sequences (in kb). We removed low-coverage contigs from the dataset if they contained <50 reads total mapped from all samples. Contigs for individuals that had only a single (i.e., unpaired) read mapped were not counted as present in that individual for number of virus and viral intensity calculations. As we did not directly enumerate viral particle counts via quantitative polymerase chain reaction (qPCR) or other methods, all quantitative descriptions hereafter use the terms “number of viruses” to describe the number of unique viral sequences (contigs) present in a sample and “intensity” to describe vRPM/kb measures for analysis purposes. Although CD-HIT-EST clustering thresholds were set to consolidate contigs sharing ≥97% nt similarity, all resulting contigs analyzed and counted in calculations for number of viruses shared < 90% nt similarity. 

We compared number of viruses, viral intensity, and CDNV-1 levels between sites using analysis of variance (ANOVA) with Tukey honestly significant difference (HSD) tests using a significance threshold of *p* < 0.05. We used linear models to compare CDNV-1 intensity levels to cumulative number of virus and viral intensity metrics in R v.4.4.0 [41]. We used a combination of multivariate methods, including variation partitioning (implemented with the varpart function of the vegan package v2.6-4 [42] in R) and nonmetric multidimensional scaling (implemented with the metaMDS function settings: distance = “bray”; center = TRUE; scale = TRUE, n = 1000 of the vegan package) to evaluate the effects of sample site, study year, and time of year (quantified by assigning each sampling event to the week of the year it occurred) on mussel virome composition. We tested the statistical significance of sample variables (with a significance threshold of *p* < 0.05) on mussel viromes using PERMANOVA, implemented in R with the adonis2 function of the vegan package (settings: permutations = 999, method = “bray”, by = “terms”). We analyzed temporal shifts and virome community stability using the codyn package v2.0.5 [43] in R. For both years of study, we calculated virus species turnover, mean rank shifts, and rate of community change for mussel viromes. Species turnover was calculated as Turnover = [(Species Gained + Species Lost)/(Total Species in both time points)]. The baseline samples from the hatchery collected at the start of each deployment served as the initial time point in the series, with changes to viromes for each site calculated at subsequent intervals.

We evaluated growth rates for all mussels placed in silos in 2020 and 2021 by calculating the difference in shell length between collection and deployment. We used the nearest U.S. Geological Survey (USGS) stream gage in the Clinch River with a temperature probe to calculate growing degree-days (GDDs), a measure of heat accumulation commonly used in agriculture to predict plant and animal development rates [44]. Stream gage data were retrieved from USGS gage stations at Kyles Ford (gage #03527620), Dungannon (gage #03524740), and Cleveland (gage #03524000) [45]. We calculated GDDs for the period of deployment for all Clinch River sites in 2020 and 2021 using the formula: GDD = ((T_max_ + T_min_)/2) − T_base_, where T_max_ and T_min_ are the daily maximum and minimum temperatures, respectively, recorded by the gages. We calculated GDDs using both T_base_ = 20 °C and T_base_ = 15 °C for further evaluation in models based on known temperature thresholds for freshwater mussel growth [46]. We also collected corresponding gage data for average flow at Clinch River sites for both years. We used USGS StreamStats [47] to calculate the watershed drainage area for each study site as a surrogate for stream size. To evaluate the relationship of environmental metrics (flow, GDDs, duration of deployment) and virus metrics (number of viruses and virus intensity) on mussel growth, we created a linear model with all metrics included and then used the stepAIC() function of the MASS package [48] in R to conduct stepwise model selection for the best combination of explanatory variables. Stream gages with temperature and flow data were not available for the NFH, MFH, or POW sites, so growth modeling was conducted for Clinch River sites only. 

## 3. Results

### 3.1. Survival and Growth Data

All mussels deployed for the 2020 study survived the full length of the study period (i.e., there was no mortality). The average length (± standard deviation (SD)) of juvenile mussels at the outset of the 2020 study was 12.94 ± 2.00 mm. The average growth was low, with average daily growth rates of 0.01–0.02 mm shell growth for all samples. The average growth for mussels deployed to sites for the full 59 days was 0.42 ± 0.20 mm at CP and 0.52 ± 0.13 mm at SYC (Table 1). The average length of the eight hatchery mussels collected at the outset of the study was 13.21 ± 1.83 mm, while the average length of the additional eight hatchery mussels collected at the end of the study was 11.9 ± 0.56 mm. During the study period, wild adult mussels were observed moribund and recently dead at the SYC study site but not at the CP site. Juvenile mussels placed in silos did not show signs of morbidity during the study period. 

Mussels placed at the five study sites during the 2021 silo study had high survival rates at two sites (100% at BEN and 100% at NFH), while three sites were affected by a flooding event during the study and experienced consequent high mortality. All mussels from one silo at the MFH site were dead upon inspection at the first retrieval date due to excess sediment accumulation and presumed smothering. An additional silo of mussels was transferred from the hatchery control cohort to the MFH site on 22 August 2021 to compensate. After the first sampling event, one of the silos at the SYC study site (containing six mussels) was lost during a flooding event, whereas a single mussel was found dead in the other silo at the second sampling event. As the Powell River site was affected by flooding and only yielded data from the first collection interval, we excluded this site from calculations of univariate statistics. Among the pheasantshell selected as the hatchery cohort for this study, we documented a 100% survival rate until the final sampling interval, at which time we observed a 50% mortality rate (*n* = 6/12 mussels) due to a temperature spike in the hatchery water supply pond (>30 °C). Details describing the number of individuals deployed to each site, survival, number of mussels collected, and details of mussels included in each sequencing pool are provided in Appendix A.

The average length (±SD) of juvenile mussels at the outset of the 2021 study was 11.4 ± 2.22 mm. Growth rates for 2021 mussels varied by site, with the highest growth values observed at the BEN site in the Clinch River (the average growth of mussels for the full 126-day deployment = 12.09 ± 1.70 mm. At the first time interval (T + 34 days), the POW mussels had the second highest growth (average = 5.46 ± 0.43 mm), but no additional growth data were available after the remaining mussels were lost. The average growth values for the full study length were similar between the SYC (7.90 ± 1.14 mm) and NFH (7.09 ± 1.17 mm) sites (Table 1). As replacement mussels were transferred to the MFH site partway through the study, we calculated growth rates separately for mussels released on the original date from those supplemented to the site on 22 August. MFH growth rates were the lowest observed of any site for the first (2.18 ± 0.60 mm) and second (2.00 ± 1.98 mm for original mussels, −0.03 ± 1.55 mm for replacement mussels) time intervals, but at the conclusion of the study, MFH mussels from the original release batch had the second highest growth rates (9.58 ± 2.36 mm) observed in the study. Mussels from the replacement silo in the MFH averaged 3.03 ± 1.77 mm of growth at the conclusion of the study. For all other sites, we observed little difference in growth for mussels from the second time interval (T + 66 days) compared to the third time interval (T + 125 days).

Model selection procedures relating average mussel growth with environmental and virus metrics (described below) in Clinch River sites yielded a final model including GDDs with a base temperature of 20 °C (*p* < 0.001), number of viruses (*p* = 0.044), watershed drainage area (*p* < 0.001), and duration of exposure (*p* = 0.177). The overall final model was significant (*p* < 0.001) with coefficient of determination (R^2^) adjusted = 0.973. Growth was positively related to GDD (estimate ± standard error 0.014 ± 0.002) but negatively related to the number of viruses (−0.008 ± 0.003) and watershed drainage area (−0.009 ± 0.001).

### 3.2. Virus Sequencing, Characterization, and Diversity

We sequenced tissues from 131 mussels combined into 27 sequencing pools by sample site and collection date. The average sequencing depth for each pool was 1,332,850 (SD ± 178,100) reads of average length 173.98 (±13.34) after quality trimming. We identified 681 viral contigs from 27 families ranging from 954 to 11,686 nt (contigs shorter than the 1000 nt cutoff occurred due to the trimming of direct terminal repeats in viruses with circular genomes) (Appendix A). Most viral contigs had low similarity to known viruses, although 50 contigs matched reference virus genomes in DIAMOND results with ≥90% similarity. Nine of these included nearly identical matches (DIAMOND similarity ≥ 97.4%) to viruses described in previous studies of North American freshwater mussel viruses [22,23]. 

The largest fraction of the mussel virome dataset belonged to the (+)ssRNA viruses (Figure 2), with particularly rich diversity in the order Picornavirales (29% of all viruses). Among these, 39% were unclassified below the order Picornavirales, 30% were members of the *Marnaviridae* family, and 17% were members of the *Dicistroviridae* family. The phylum Cressdnaviricota comprised 28% of all viruses, although the majority of these (83%) were not classified below the phylum level. Other common families included the *Parvoviridae*, *Nodaviridae*, and *Tombusviridae* (6%, 5%, and 4% of all viruses, respectively). “Viruses, unclassified” comprised 9% of all contigs observed.

### 3.3. Distribution of Viruses over Study Sites and Sample Periods

We observed 230 viruses in the hatchery, 20 of which were exclusive to the hatchery samples (Appendix A). These 20 included 10 ssDNA viruses, 4 picorna-like viruses, 3 members of the Tolivirales, 2 nodaviruses, and a picobirnavirus. We observed 451 viruses exclusively in samples from river sites (i.e., never found in the hatchery) (Appendix A). These 451 included the Pisuviricota (n = 149), almost all of which (131) belonged to the Picornavirales, including 36 *Marnaviridae* and 23 *Dicistroviridae*. ssDNA viruses of the Phylum Cressdnaviricota were the next most abundant group found exclusively in rivers, with 126 viruses. Approximately 20% of the viruses found only in rivers were unclassified, and other commonly observed groups included the Kitriniviricota and Coassaviricota (all of which were members of the *Parvoviridae*). Five families were never observed in the hatchery: *Bacilladnaviridae*, *Partitiviridae*, *Permutotetraviridae*, *Secoviridae*, and *Tymoviridae*. Each of these five families was represented by a single virus. The Clinch River sites (CP, BEN, and SYC) yielded 88 viruses that were never observed in the hatchery or other rivers (Appendix A). However, most of these viruses were observed across multiple Clinch River sites, with only 1, 2, and 8 viruses found exclusively at the CP, BEN, and SYC sites, respectively (Appendix A). At other sites, 24 viruses were unique to the MFH, 20 to the NFH, and 2 to the POW (Appendix A). We found 37 viruses exclusively in the first year of the study and 170 exclusively in the second year, while 474 viruses were observed in both years (Appendix A). 

We found three viruses present in all 27 sequencing pools, including CDNV-1, clidnapec virus 9 (a novel densovirus), and clictolig virus 3, an ssDNA virus previously identified from wild adults of the congeneric mucket (*Actinonaias ligamentina*) from the Clinch River sampled in 2018 [23] (Appendix A). Clidnapec virus 9 and clictolig virus 3 had similar and stable levels of virus intensity between the hatchery samples (average clidnapec virus 9 vRPM/kb 1.80 ± 0.42 SD) and wild site samples (average clidnapec virus 9 vRPM/kb 1.59 ± 0.55 SD, *t*-test: *t*-value = 0.90, *p* = 0.38, df = 5) over time. In addition to the three viruses found in all 27 sample pools, we observed seven DNA viruses in all 21 sample pools from the wild sites. Among these, five viruses (two members of the family Parvoviridae and three small circular DNA viruses) were never observed in the hatchery, and two viruses (one member of the *Parvoviridae* and one small circular DNA virus) were observed in one of six hatchery sample pools at low levels. All of these viruses were novel in that they had low sequence similarity (highest = 78.0% nt identity over 37% genome length) to nearest matches in GenBank.

CDNV-1 was present in all sequencing pools but was found at significantly higher intensity in the wild sites (mean vRPM/kb ± SD = 4.08 ± 0.83) than in hatchery samples (1.39 ± 0.43) (two-tailed *t*-test, *p* < 0.001). ANOVA with Tukey HSD tests revealed that the mean viral intensity for CDNV-1 was significantly lower in the hatchery than at all wild study sites (Appendix A). Among the wild sites, CDNV-1 intensity was significantly lower at SYC (2.77 ± 0.78) than BEN (4.18 ± 0.04, *p* adj. = 0.013) and NFH (4.19 ± 0.42, *p* adj. = 0.013). No other differences were significant among the wild sites for CDNV-1 intensity. CDNV-1 levels consistently exhibited rapid ~1–3 log increases compared to the baseline hatchery samples by the first sampling interval in both study years. CDNV-1 had the highest intensity observed in the study (vRPM/kb = 5.31 in pool CS27 from the NFH site) and was the virus with the highest intensity observed in 7 of 21 sequencing pools from the wild sites. Linear models showed a significant positive relationship between CDNV-1 concentration and the number of viruses (R^2^ = 0.57, *p* < 0.001). Models revealed a significant positive relationship between CDNV-1 concentration and cumulative virus intensity (R^2^ = 0.47, *p* < 0.001) when CDNV-1 intensity was included in the cumulative calculation, but no significant relationship remained when virus intensity was calculated for all viruses except for CDNV-1 (R^2^ = 0.14, *p* = 0.052).

### 3.4. Virome Metrics

The average number of viruses in the hatchery mussel sample pools (mean ± SD = 76.67 ± 39.74) was significantly lower than in all of the silo study sites with the exception of NFH. No differences in the number of viruses were significant between any of the silo study sites (ANOVA with Tukey HSD) (Figure 3; Appendix A). The average virus intensity (vRPM/kb) was not significantly different between most study sites, with the exception that MFH (2.00 ± 0.21) was significantly higher than AWCC (1.03 ± 0.41; *p* = 0.039) and SYC (1.06 ± 0.56; *p* = 0.041) (ANOVA with Tukey HSD, *p*-values adjusted for multiple comparisons). Across both years, the average number of viruses at all of the wild sites was 2.87× (±1.12 SD) higher at the first sampling interval than at the prior measurement at the hatchery. In the second year when hatchery and silo samples were collected contemporaneously, the number of viruses in the wild samples was 4.18× (±2.01 SD) higher than time-matched hatchery samples. The number of viruses continued to increase after the first sampling interval for the wild sites, increasing by an average of 1.72× ± 0.70 from the first to the last sampling interval for all of the wild sites. The number of viruses in the hatchery samples increased by 1.78× and 2.21× from the outset to the conclusion of the study period in 2020 and 2021, respectively, but consistently remained lower than the time-matched wild samples.

Non-metric multidimensional scaling (NMDS) ordination indicated a distinct separation of hatchery mussel viromes from those from the river sites. Viromes in the river sites generally clustered by the sample site. All of the sites in both years had the largest shifts in virome composition (compared to the baseline hatchery samples) at the first sampling interval, with much smaller shifts thereafter (Figure 4). The PERMANOVA results showed that study site (*p* = 0.001), the time of year (*p* = 0.001), and study year (*p* = 0.019) all significantly influenced mussel virome composition (Appendix A). Variation partitioning models indicated that among these variables, the sample site was the most important factor, explaining 41.2% of variation in mussel virome composition (R^2^ = 0.412 after multiple comparison adjustment), with 34.0% of variation uniquely attributable to the sample site in the model (Appendix A). Most of the explanatory power of the other variables was in the fraction of shared variance explained, as the study year explained 10.6% of virome variation but just 1.9% of unique variation. The time of year had the lowest proportion of virome variation explained (6.6%) but had a slightly higher unique proportion of variation explained (3.5%) when compared to the study year. 

In 2020, the CP and SYC sites exhibited similar trends over time in virome composition and stability. The number of viruses increased from 64 in the hatchery baseline samples to 245 at CP and 209 at SYC for the first sample interval (t + 14 days). The number of viruses continued to increase over time, with SYC reaching peak viral numbers at the second sampling interval on October 7 (n = 315) before decreasing to levels similar to the first interval in the third and fourth intervals. The CP site had a steadily increasing number of viruses through the third sampling interval, peaking at 388, and remained high for the fourth and final sampling interval (n = 373). The hatchery virus numbers approximately doubled from the outset of the study in September (n = 64) to the conclusion in November (n = 114).

The total turnover and mean rank shift were highest at the first sampling interval after deployment and steadily decreased in subsequent sampling events for both sites (Figure 5, second and third rows). This pattern was driven largely by the appearance rate metric, which was high at the first sample interval (0.76 for CP and 0.74 for SYC) and steadily declined in subsequent intervals. Disappearance rates for CP and SYC in 2020 were lower, ranging from 0.06 to 0.36 (mean = 0.17 ± 0.10 SD). The rate of virome community change for both sites was similar and showed a positive slope (Figure 5, bottom), indicating that viromes for each site were increasingly dissimilar from previous values over time.

In 2021, the mussels accumulated a large number of viruses upon release to rivers, followed by an increasing accumulation as they spent more time in the river. The initial hatchery samples collected in June had similar, relatively low, virus numbers (n = 62) compared to the previous year. Virus numbers in the hatchery mussels remained similar in the July (n = 39) and August (n = 44) samples before a large increase to the final samples collected in October (n = 137). On the first sampling interval (T + 31 days), while the hatchery virus numbers decreased to 39, the wild-site virus numbers increased to levels ranging from 101 (SYC) to 273 (MFH). The virus numbers steadily increased in subsequent sampling intervals, and all of the sites (including the hatchery) had the highest viral numbers at the final (third) sampling interval on October 26th (T + 124–126 days).

The turnover rates for 2021 samples exhibited similar trends to those observed in 2020. The total turnover was highest at the first sample interval for all of the wild sites, which was again driven by high initial appearance rates (Figure 6, second row). The appearance rates in the wild sites (mean = 0.57 ± 0.24 SD) declined in subsequent sample intervals, while the disappearance rates were lower and more stable (mean = 0.20 ± 0.12 SD). The opposite pattern was observed in the hatchery (AWCC), with the turnover rates steadily increasing throughout the study, driven by a consistently increasing appearance rate and decreasing disappearance rate. Across all of the sample sites in both years, the virus numbers at the first sampling interval increased by an average of 2.87× (±1.12 SD) compared to the previous sample collected from the hatchery. The rates of virome community change over time in 2021 were larger and more rapid in the MFH and NFH sites compared to the hatchery or either of the Clinch River sites (BEN and SYC) (Figure 6, bottom). 

## 4. Discussion

In a longitudinal study of freshwater mussels, we documented a large increase in virus infection after the mussels were deployed from the hatchery into silos in the wild. The number of viruses increased substantially by the first sampling interval at the wild sites and then continued to climb throughout the study period, whereas the hatchery samples consistently had much lower numbers of viruses with a small increase at the final sampling interval in both years. Mussel growth was driven almost entirely by water temperature, although we observed a significant negative relationship between mussel growth and the number of viruses. CDNV-1, which we previously identified as associated with mussel mortality in the Clinch River [22], was found at low intensity in all hatchery samples but at much higher intensity in all of the wild site samples. These trends were modified by site-specific differences within and between the rivers. Overall, our results indicate that mussels released from hatcheries for restoration purposes experience rapid and continuous increases in virus numbers upon release to river sites, and these increases are moderated by site-specific patterns that may warrant further consideration for restoration planning.

We identified 681 viruses from 27 tissue pools of juvenile pheasantshell collected from seven sites in the UTRB across 2 study years. Most viruses were previously uncharacterized and were most closely related to viruses infecting invertebrates or found in water samples [49,50]. These results are consistent with a previous investigation of freshwater mussel viromes from sites in the western United States, where high species richness and diversity were found along with similar virome composition, comprising mostly (+)ssRNA viruses, followed by small circular DNA viruses and members of the *Parvoviridae* [32]. The overall virome composition we observed is similar to that of many previous studies of marine and freshwater viromes, which have consistently documented that unclassified members of the *Picornavirales* tend to be the most abundant viruses observed [51]. Given that freshwater mussels are filter feeders and their virome profile is similar to that of many previous studies of virioplankton in marine and freshwater environments (e.g., [52]), it is tempting to hypothesize that many of these viruses are of dietary origin. However, our data indicate that the peak number of viruses in mussels occurs during periods of reduced metabolism and growth, corresponding to colder temperatures and lower river productivity. This observation indicates that virus prevalence in freshwater mussels is not simply a byproduct of filter feeding. It is also possible that the colder temperatures and corresponding reduced metabolism in mussels result in slower rates of virus degradation, causing an increase in the number of observed viruses. 

We documented 20 viruses only in the hatchery mussels but nearly 23 times more viruses (451) only in the mussels released to the wild sites. These were mainly individual members of the Phyla Cressdnaviricota, Kitrinoviricota, and Pisuviricota. Among the 451 viruses observed for the mussels exclusively in the river sites, the most common group was the Pisuviricota, followed by ssDNA viruses of the Phylum Cressdnaviricota, unclassified viruses, and smaller numbers of other groups, including Kitriniviricota and Coassaviricota. Viruses unique to individual sites and/or rivers were generally distributed in the same proportions as in the larger dataset. The number of viruses per sample was consistently much higher in this study than in previous studies of wild adult mussel viromes, which include Clinch River pheasantshell [22], congeneric mucket from the Clinch River and two other Midwestern populations [23], and two species of freshwater mussels from the western United States during MMEs [32]. These differences may be due in part to the tissues analyzed, as our previous studies analyzed hemolymph, which is an acellular liquid, rather than solid organs. Also, sequences in this study were from solid tissues of multiple individuals pooled by location and time, whereas sequences from our previous studies were from individual mussels.

The overall numbers of viruses were significantly lower in the hatchery than in most of the river sites, regardless of the time of year, temperature, duration of deployment, or sample site, but did not significantly differ among any of the river sites. The sample site was by far the most influential variable shaping virome composition, whereas the study year and time of year had smaller—but still significant—effects on virome composition. The MFH site had the highest number of viruses and the highest viral intensity values and was the only site with viral intensity values significantly different from other sites in the study. Virome turnover metrics indicated that the differences between the MFH and other sites occurred rapidly, as the divergence occurred by the first sampling interval. Thereafter, the rates of virus number increase and turnover decrease were similar between the MFH and other sites; thus, the significant differences we observed throughout the study period were driven by the rapid shift in the early portion of the deployment. Our tissue pooling approach facilitated efficient sequencing and virome characterization but resulted in a relatively low sample size per site. Finer-scale differences among sites might be revealed by future work comparing viromes of individual mussels across spatial and temporal gradients (e.g., among seasons). One potential explanation for the reduced number of viruses observed in the hatchery is that the simplified ecology of the captive setting comprises a more homogenous virome compared to the complexity of a natural river ecosystem. Alternatively, the stress of transfer from the stable, protected hatchery environment to the river sites might result in decreased immune competence and facilitate increased susceptibility to virus infection.

In a previous study of the ongoing Clinch River mussel MME, a significant relationship was found between moribund pheasantshell and CDNV-1 based on both prevalence and virus intensity [22]. In this study, CDNV-1 was present in all sample pools, but virus levels increased and remained high when the mussels were deployed to the wild sites. In contrast to previous results, we observed significantly lower levels of CDNV-1 at the SYC site, where the mortality of wild adult pheasantshell was ongoing, when compared with the far upstream BEN control site, where no mortality of wild adults was apparent. CDNV-1 was the most prevalent virus in the study (along with two others—clidnapec virus 9 and clictolig virus 3, found in all 27 sequencing pools). CDNV-1 intensity was consistently several orders of magnitude lower in hatchery samples than in time-matched silo samples placed in rivers. The observed increases in CDNV-1 intensity were rapid, as they were observed at the first sampling interval (2 weeks after deployment) in 2020. CDNV-1 levels appeared high but relatively stable over time at each site, and they did not vary with observed fluctuations in mussel growth and water temperature. The hatchery juveniles were CDNV-1 positive but did not exhibit signs of morbidity or mortality when exposed to the ongoing MME at SYC. These findings indicate that CDNV-1 could be affecting mussel health in a context-dependent manner or that CDNV-1 could be responding to the physiological state of the mussel but not causing health effects itself.

Our data provide evidence that restoration efforts may be influenced by site-specific variation not only via abiotic variables classically thought to control mussel restoration success (e.g., flow, temperature, and physical habitat) but also site-dependent factors shaping virome and microbiome composition. To our knowledge, this is the first study of (1) juvenile freshwater mussel viromes, (2) freshwater mussel viromes over time using comparable cohorts, and (3) the effect of restoration practices on freshwater mussel viromes. The abundance and diversity of novel viruses characterized in this and other recent studies of aquatic [50] and invertebrate [49] viromes make it difficult to examine the significance of any particular virus (or group of viruses). However, even given the remarkable diversity of invertebrate viromes, our study shows that it is possible to identify factors shaping invertebrate virus community composition and abundance. To the extent that viruses and other infectious agents pose a health risk to hatchery-reared mussels released into the wild (and the magnitude of this risk is yet to be determined), the consideration of strategies to manage this risk may be warranted in captive propagation efforts. For example, mussels might be exposed gradually and in a controlled manner to river water prior to full release (acclimatization). If individual infectious agents are identified that cause disease in mussels, then controlled exposure, vaccination, or other strategies could be considered. Given the imperiled status of mussel taxa being reared in captivity, any measures to increase post-release survival would be important.

## 5. Conclusions

We documented 681 viruses in propagated juvenile freshwater mussels from a conservation hatchery and surrounding wild restoration sites in four rivers of the UTRB. We documented significantly lower numbers of viruses in the hatchery setting, as well as rapid virome shifts and increased numbers of viruses in juvenile mussels upon exposure to wild sites. Virome composition in silo-deployed mussels varied most strongly based on the sample site, with the difference between the hatchery and wild sites larger than that observed between the individual wild sites.

Freshwater mussels are among the world’s most imperiled organisms. As MMEs, habitat destruction, pollution, climate change, and invasive species continue to erode remaining mussel populations, conservation intervention is increasingly beneficial. The primary method in mussel conservation involves aquaculture to propagate animals in captivity for the augmentation and restoration of affected populations. To avoid inadvertent harm to existing populations, as is routinely documented in other large-scale aquaculture efforts (e.g., marine bivalves and finfish), it is important to characterize and understand the microbial symbionts and pathogens of freshwater mussels propagated and released to wild sites. The mussels grown and held in the hatchery environment appear to have a substantially lower abundance and diversity of aquatic viruses. When released to the restoration sites, the mussel viromes exhibit rapid shifts in abundance and diversity. Captive propagation efforts could consider strategies such as acclimatization to mitigate possible risks of the rapid exposure of hatchery-reared mussels into the wild.

## Figures and Tables

**Figure 1 animals-14-02531-f001:**
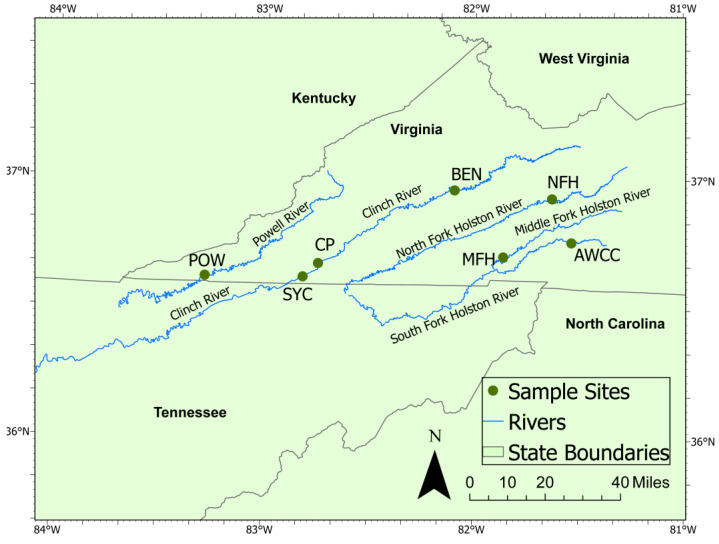
Map of sampling sites and the Aquatic Wildlife Conservation Center (AWCC) mussel hatchery used in the 2020–2021 mussel silo study. Abbreviations used in text are CP (Clinchport, Clinch River, mile 213.2); SYC (Sycamore Island, Clinch River, mile 206.9); POW (Powell River, mile 120.2), BEN (Clinch River, mile 278.1), NFH (North Fork Holston River, mile 97.2), and MFH site (Middle Fork Holston River, mile 8.3).

**Figure 2 animals-14-02531-f002:**
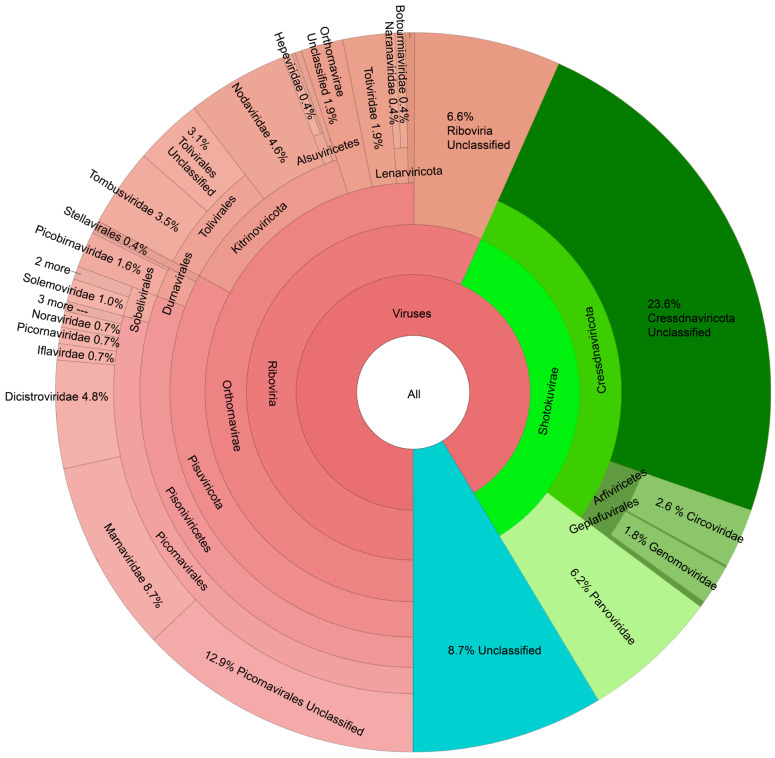
KRONA diagram showing taxonomy of the 681 viruses observed in juvenile freshwater mussels deployed in sites from the Upper Tennessee River Basin. Groups representing <0.4% of all viruses are collapsed and shown with cross-hatch shading.

**Figure 3 animals-14-02531-f003:**
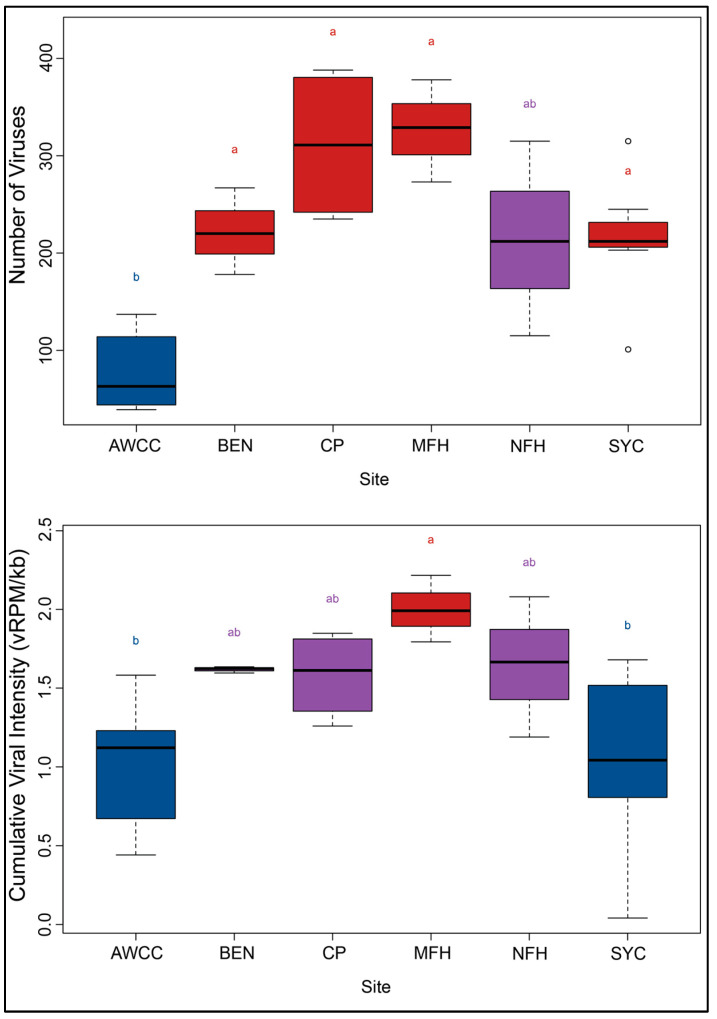
Boxplots showing average number of viruses by sample site (**Above**) and cumulative viral intensity (**Below**). Boxes depict median and first and third quartiles for each group. Upper whiskers represent the smaller of either the maximum observed value of 1.5× the 3rd quartile. Lower whiskers represent the larger of either the minimum observed value or 1.5× the 1st quartile subtracted from the value of the 1st quartile. Letters and colors depict results from Tukey HSD analysis. Boxes with the same letters and colors are not statistically different (i.e., adjusted *p*-value ≥ 0.05).

**Figure 4 animals-14-02531-f004:**
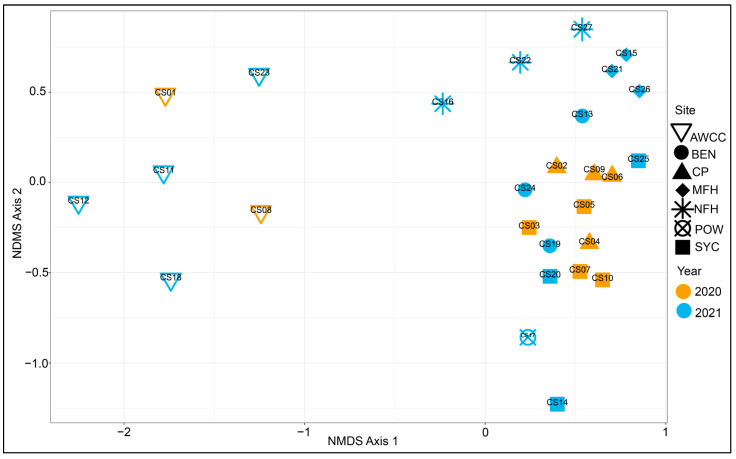
Non-metric multidimensional scaling (NMDS) plot showing virome results for all sequencing pools included in the Upper Tennessee River Basin mussel silo study. For both years, there was a clear separation of viromes in the mussels held at the hatchery (hollow inverted triangles; Aquatic Wildlife Conservation Center, AWCC) from those deployed to river sites (all other shapes). River viromes clustered largely by sample site. All three sample sites within the Clinch River (SYC, CP, and BEN) largely overlapped, while sites from the Middle Fork Holston River (MFH), North Fork Holston River (NFH), and Powell River (POW) did not overlap with samples from any other rivers.

**Figure 5 animals-14-02531-f005:**
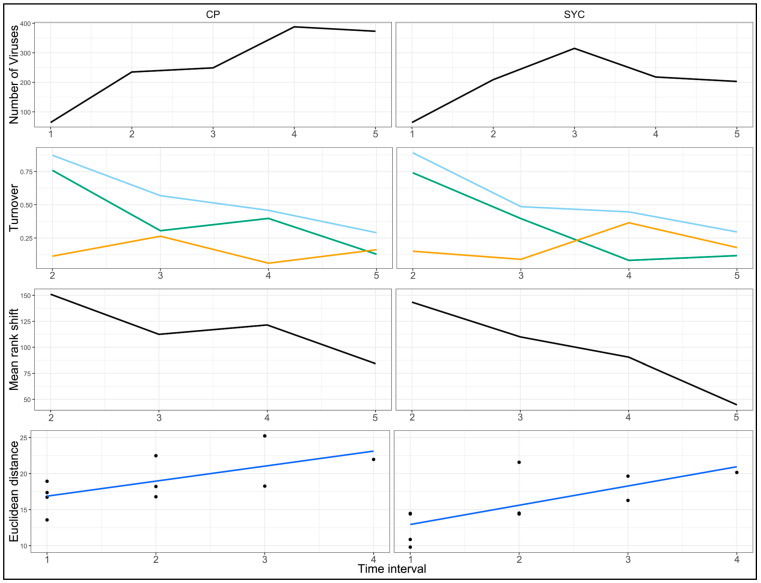
Changes in freshwater mussel virome composition at two Clinch River sites over 4 sampling intervals in 2020. Top row: number of viruses. Second row: virome turnover plots showing total turnover rate (light blue line), disappearance rate (orange line), and appearance rate (green line). Third row: mean rank shifts representing the degree of virus abundance reordering between time points. Fourth row: rate of virome community change. Note that the top plot includes five values, including the baseline value from the outset of the study, while all other plots contain four values, representing observed changes from the previous interval.

**Figure 6 animals-14-02531-f006:**
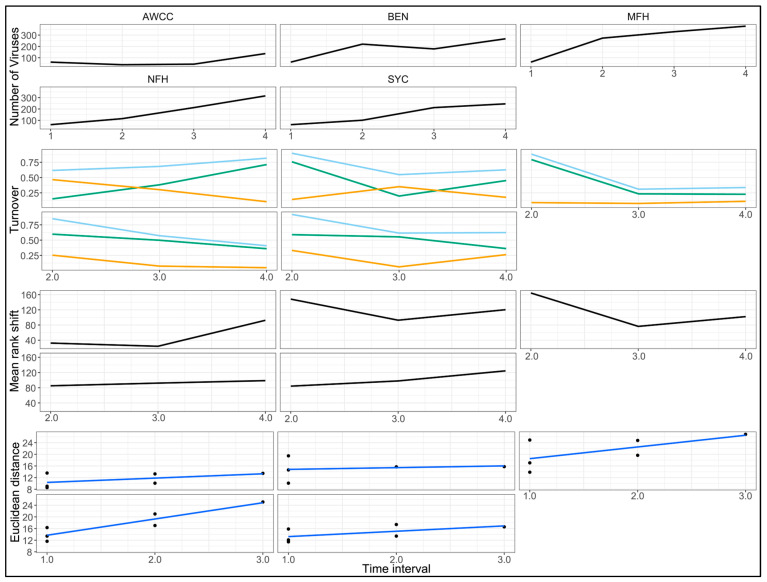
Changes in freshwater mussel virome composition at 4 river sites (Clinch River sites BEN and SYC, Middle Fork Holston River (MFH), North Fork Holston River (NFH)), and the Aquatic Wildlife Conservation Center (AWCC) hatchery over 3 sampling intervals in 2021. Top row: number of viruses. Second row: virome turnover plots showing total turnover rate (light blue line), disappearance rate (orange line), and appearance rate (green line). Third row: mean rank shifts representing the degree of virus abundance reordering between time points. Fourth row: rate of virome community change. Note that the top plot includes four values, including the baseline value from the outset of the study, while all other plots contain three values, representing observed changes from the previous interval.

**Table 1 animals-14-02531-t001:** Sequencing pools, number of viruses, viral intensity, deployment dates, and growth values for juvenile freshwater mussels included in the Upper Tennessee River Basin virome study. Growth rates were not tracked for mussels held in the hatchery period in either year. Asterisks indicate growth rates for mussels deployed to the MFH site on 22 August 2021 to replace those lost due to heavy siltation in silos. SD = standard deviation.

River	Site Code	Sample	Number of Viruses	Viral Intensity (VRPM/kb)	CDNV-1 (VRPM/kb)	Deploy Date	Collection Date	Days Deployed	Average Growth (mm) ± SD
Hatchery	AWCC	CS01	64	0.67	0.33	n/a	9/9/2020	n/a	n/a
Hatchery	AWCC	CS08	114	1.58	0.82	n/a	11/4/2020	n/a	n/a
Clinch	CP	CS02	235	1.26	2.65	9/9/2020	9/22/2020	13	0.28 ± 0.13
Clinch	CP	CS04	249	1.78	3.88	9/9/2020	10/7/2020	28	0.44 ± 0.34
Clinch	CP	CS06	388	1.85	3.71	9/9/2020	10/20/2020	41	0.47 ± 0.32
Clinch	CP	CS09	373	1.45	3.42	9/9/2020	11/6/2020	58	0.42 ± 0.20
Clinch	SYC	CS03	209	0.75	2.68	9/9/2020	9/22/2020	13	0.25 ± 0.13
Clinch	SYC	CS05	315	1.59	3.69	9/9/2020	10/7/2020	28	0.34 ± 0.18
Clinch	SYC	CS07	218	1.68	3.48	9/9/2020	10/20/2020	41	0.40 ± 0.24
Clinch	SYC	CS10	203	1.45	3.22	9/9/2020	11/6/2020	58	0.52 ± 0.13
Hatchery	AWCC	CS11	62	1.08	0.77	n/a	6/25/2021	n/a	n/a
Hatchery	AWCC	CS12	39	1.23	0.53	n/a	7/26/2021	n/a	n/a
Hatchery	AWCC	CS18	44	0.44	0.79	n/a	8/27/2021	n/a	n/a
Hatchery	AWCC	CS23	137	1.16	1.43	n/a	10/25/2021	n/a	n/a
Clinch	BEN	CS13	220	1.63	4.15	6/22/2021	7/26/2021	34	6.50 ± 0.74
Clinch	BEN	CS19	178	1.62	4.22	6/22/2021	8/27/2021	66	11.32 ± 1.79
Clinch	BEN	CS24	267	1.60	4.18	6/22/2021	10/26/2021	126	12.09 ± 1.70
Clinch	SYC	CS14	101	0.04	1.58	6/23/2021	7/26/2021	33	3.66 ± 0.91
Clinch	SYC	CS20	212	1.04	2.82	6/23/2021	8/27/2021	65	6.18 ± 0.73
Clinch	SYC	CS25	245	0.87	1.94	6/23/2021	10/25/2021	124	7.90 ± 1.14
Middle Fork Holston	MFH	CS15	273	1.99	3.16	6/21/2021	7/26/2021	35	2.18 ± 0.60
Middle Fork Holston	MFH	CS21	329	1.79	3.64	6/21/2021	8/27/2021	67	2.00 ± 1.98 (−0.03 ± 1.55) *
Middle Fork Holston	MFH	CS26	378	2.22	3.92	6/21/2021	10/25/2021	126	9.58 ± 2.36 (3.03 ± 1.77) *
North Fork Holston	NFH	CS16	115	1.19	3.78	6/21/2021	7/26/2021	35	3.56 ± 0.68
North Fork Holston	NFH	CS22	212	1.67	4.18	6/21/2021	8/27/2021	67	7.00 ± 0.89
North Fork Holston	NFH	CS27	315	2.08	4.62	6/21/2021	10/25/2021	126	7.09 ± 1.17
Powell	POW	CS17	107	1.20	2.04	6/23/2021	7/26/2021	34	5.46 ± 0.43

## Data Availability

The sequence reads generated in this study are available at the NCBI Sequence Read Archive (SRA) database under BioProject PRJNA1126263, accession numbers SRX25030521–SRX25030547. Viral sequences described in this study have been deposited in GenBank under accession numbers PQ161420-PQ162100. Details for all individual mussel samples, viruses described in this study, viral number, viral read depths, and statistical tests are provided in Appendix A.

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
