# Peer review of "Freshwater Mussel Viromes Increase Rapidly in Diversity and Abundance when Hosts Are Released from Captivity into the Wild"

_animals, 2024, doi:10.3390/ani14172531_

Round 1

Reviewer 1 Report

Comments and Suggestions for Authors

Present study investigated the viromes of freshwater mussel in different sites of the Upper Tennessee River Basin in Virginia. Virus abundances between multiple restoration sites and the hatchery were also compared. The conclusions can be supported by the results. But the manuscript needs some revisions before it could be accepted.

 1. The “Number of viruses” was used in the MS to describe the virus abundance in different samples/sites. But it’s unclear how to define the “number of viruses”.  If one contig means one virus, how to process the contigs with similarities? In other words, how much difference is required for two contigs to be considered distinct? Could you please clarify?

2. Tables and Figures are not properly formatted to make their data uncomprehensible. E.g. The horizontal lines not showing in Table 1. The font in the figure 2 is confused in the direction   

3. The CDNV-1 levels were compared and highlighted, so some results including the CDNV-1 levels should be presented in table 1. In addition, Clinch densovirus 1 (CDNV-1) does not need to be repeated many times, or can be replaced by CDNV-1.

4. The family name of the viruses should be written in italics (e.g. line 386-387).

5. About the background in the Indroduction, the authors neglected some recent and directly related references. So, it is recommended to read and cite the relevant literature (Zhang QY, Ke F, Gui L, Zhao Z. 2022. Recent insights into aquatic viruses: emerging and reemerging pathogens, molecular features, biological effects, and novel investigative approaches. Water Biology and Security, 1 (4):100062)

Author Response

Present study investigated the viromes of freshwater mussel in different sites of the Upper Tennessee River Basin in Virginia. Virus abundances between multiple restoration sites and the hatchery were also compared. The conclusions can be supported by the results. But the manuscript needs some revisions before it could be accepted.

Comment 1. The “Number of viruses” was used in the MS to describe the virus abundance in different samples/sites. But it’s unclear how to define the “number of viruses”.  If one contig means one virus, how to process the contigs with similarities? In other words, how much difference is required for two contigs to be considered distinct? Could you please clarify?

We appreciate the reviewer raising this important point, as we had numerous internal discussions during the manuscript draft and revision stages attempting to make sure that there is no ambiguity about this point. As described earlier in the methods sections, the CD-HIT-EST threshold was set to cluster sequences with ≥97% nucleotide identity, but in practice, our resulting contigs were based on much higher differences. We have added text at lines 259-261 to clarify the similarity of contigs in the dataset, stating:

While CD-HIT-EST clustering thresholds were set to consolidate contigs sharing ≥97% nt similarity, all resulting contigs analyzed and counted in calculations for number of viruses shared <90% nt similarity.

Comment 2. Tables and Figures are not properly formatted to make their data uncomprehensible. E.g. The horizontal lines not showing in Table 1. The font in the figure 2 is confused in the direction   

Response: We appreciate the reviewer’s assistance in noting these details that will improve the presentation and readability of the manuscript. We have modified Table 1 per MDPI formatting guidelines to create additional border lines separating the header line and delineating the bottom of the table (at line 331). We have also fixed the spacing which caused the Table 1 header to appear on a separate page from the rest of the table in the original submission. We will confer with the Editors to determine if additional lines within the table should be added to more effectively delineate the contents while following formatting guidelines.

Comment 3. The CDNV-1 levels were compared and highlighted, so some results including the CDNV-1 levels should be presented in table 1. In addition, “Clinch densovirus 1 (CDNV-1)” does not need to be repeated many times, or can be replaced by “CDNV-1”.

Response: We thank the reviewer for these suggestions. We have removed superfluous instances of both the full virus name and abbreviation at lines 119 and 399. We have modified table 1 to add CDNV-1 levels as suggested.

Comment The family name of the viruses should be written in italics (e.g. line 386-387).

Response: We appreciate the reviewer catching this mistake. We have reviewed the manuscript and corrected all instances of the family names that did not appear in italics.

Comment 5. About the background in the Indroduction, the authors neglected some recent and directly related references. So, it is recommended to read and cite the relevant literature (Zhang QY, Ke F, Gui L, Zhao Z. 2022. Recent insights into aquatic viruses: emerging and reemerging pathogens, molecular features, biological effects, and novel investigative approaches. Water Biology and Security, 1 (4):100062)

Response: This is an important point, and we thank the reviewer for raising it. We have included this reference in our introduction during the discussion of viruses affecting aquaculture and wild aquatic animals at line 84.

Reviewer 2 Report

Comments and Suggestions for Authors

The manuscript submitted by Richard et al. presents a very interesting issue that is a continuation of the authors' earlier research with many innovative elements.

It may be an example for similar research undertaken in the future, especially in the case of attempts to restore species, e.g. endangered bivalve.

In my opinion, the element that should be emphasized in the Discussion and Conclusions is the explanation of why the diversity of viruses is higher after being placed in the river environment than in the hatchery. Of course, the main reason is quite obvious to me, but I would like the authors to try to indicate the reasons in the Conclusions. After all, it is known that the microbiome of filtered water from the pond in the hatchery and current river waters will be different. Additionally, the decrease in immunity after transferring juvenile organisms to river waters causes a decrease in immunity, which certainly affects the ease of virus penetration into the body. Or do the authors have another explanation they could provide?

In addition, please clarify and correct the issues in the following lines:

- line 40: federally listed? Please explain what this is about, it is incomprehensible to global readers who do not come from the US.

- line 156-157: "Mussels were held in concrete silos, which 156 allow river water to flow through a mesh screen while preventing escape (Supplemental 157 Figure S1)". Please indicate where Figure S1 is located, I did not find it in the Supplementary files.

- line 327: Please convert Table 1 according to publisher requirements (font etc.)

- line 457: indicate the location of Supplementary Figure S2, similar to Figure S1

- line 672: Please provide Genbank accession numbers for viral sequences

Author Response

The manuscript submitted by Richard et al. presents a very interesting issue that is a continuation of the authors' earlier research with many innovative elements.

It may be an example for similar research undertaken in the future, especially in the case of attempts to restore species, e.g. endangered bivalve.

In my opinion, the element that should be emphasized in the Discussion and Conclusions is the explanation of why the diversity of viruses is higher after being placed in the river environment than in the hatchery. Of course, the main reason is quite obvious to me, but I would like the authors to try to indicate the reasons in the Conclusions. After all, it is known that the microbiome of filtered water from the pond in the hatchery and current river waters will be different. Additionally, the decrease in immunity after transferring juvenile organisms to river waters causes a decrease in immunity, which certainly affects the ease of virus penetration into the body. Or do the authors have another explanation they could provide?

Response: We appreciate the reviewers insight on this point, and we agree completely. We have added additional discussion from lines 589-594 to further discuss the potential reasons for the observed patterns of decreased virus abundance in the hatchery setting compared to the river sites.

In addition, please clarify and correct the issues in the following lines:

Comment 1 - line 40: federally listed? Please explain what this is about, it is incomprehensible to global readers who do not come from the US.

Response: We appreciate this important point from the reviewer. We have revised the text to indicate that we refer to species listed as threatened and endangered under the U.S. Endangered Species Act. At lines 40-41.

Comment  2 - line 156-157: "Mussels were held in concrete silos, which 156 allow river water to flow through a mesh screen while preventing escape (Supplemental 157 Figure S1)". Please indicate where Figure S1 is located, I did not find it in the Supplementary files.

Response: Thank you for noting this omission. We have added an additional photo (Supplemental Figure S2) that depicts the interior chamber of the mussel silos. We have also added the Supplemental Figure captions at lines 657-665 and will ensure that these supplements are uploaded with the revised manucripts.

Comment  3 - line 327: Please convert Table 1 according to publisher requirements (font etc.)

Response: We thank the reviewer for noting this formatting discrepancy. We have corrected the table meet the publishers formatting guidelines.

Comment  4 - line 457: indicate the location of Supplementary Figure S2, similar to Figure S1

Response: We again thank the reviewer for catching our omission of the supplementary figures. The supplementary figures also include Figure S3 (previously Figure S2), depicting the results of the variation partitioning analysis. The corresponding caption has been added at lines 657-665.

Comment  5 - line 672: Please provide Genbank accession numbers for viral sequences

Response: We have replaced the previous placeholder with the GenBank accession numbers PQ161420-PQ162100.

Reviewer 3 Report

Comments and Suggestions for Authors

This paper compares the virome in relation to the unexplained mass mortality of freshwater mussels by experimentally rearing hatchery-produced juveniles at multiple river sites. Although no mass mortality occurred during the study period, the regional and temporal changes in the virome were analyzed from multiple perspectives, yielding very interesting results. The methods are appropriate, and the discussion is based on the results. There are no major issues with the publication of this paper. The following are detailed remarks.

Please add the scientific name of the Pheasantshell used in the study.

Figure 1: The location of NFH is missing.

Line 222: Please briefly explain the in-house database because it's unclear how Eukaryotic reads were excluded. Does it contain genomic data for the target organisms?

Line 305: SI test sites are not listed in Figure 1.

Line 547: The authors suggest that since the virus levels are highest during periods of low water temperature, when the metabolism and growth of mussels are reduced, the viruses detected in the mussels are not merely a byproduct of filter feeding. However, could it not be possible that during periods of low water temperature, the degradation and disappearance of viruses in the water are less than during periods of high-water temperature, leading to an increase in virus accumulation in the water and, as a result, an increase in the amount of virus detected in mussels as a byproduct of filter feeding?

Author Response

This paper compares the virome in relation to the unexplained mass mortality of freshwater mussels by experimentally rearing hatchery-produced juveniles at multiple river sites. Although no mass mortality occurred during the study period, the regional and temporal changes in the virome were analyzed from multiple perspectives, yielding very interesting results. The methods are appropriate, and the discussion is based on the results. There are no major issues with the publication of this paper. The following are detailed remarks.

Comment 1: Please add the scientific name of the Pheasantshell used in the study.

Response: The scientific name for pheasantshell (Actinonaias pectorosa) is placed with the first instance of the common name pheasantshell at line 101 of the revised manuscript.

Comment 2: Figure 1: The location of NFH is missing.

Response: Thank you for catching this error! The label was somehow lost during the conversion between image formats. We have replaced Figure 1 with a corrected version that includes the label for NFH.

Comment 3: Line 222: Please briefly explain the in-house database because it's unclear how Eukaryotic reads were excluded. Does it contain genomic data for the target organisms?

Response: We have added text to lines 223-225 providing additional details about the in-house database, which includes known contaminants, the NCBI UniVec database, ribosomal RNA sequences, and low-complexity repeat regions (e.g., long stretches of sequences with a single nucleotide or dinucleotide repeats).

Comment 4: Line 305: SI test sites are not listed in Figure 1.

Response: Thank you for catching this important detail! In an original draft of the paper, we had referred to the Sycamore Island study site with the designation “SI” rather than “SYC” and this was an error that was not updated. We have carefully checked the manuscript and replaced the code “SI” with “SYC” at lines 309, 392, and 395.

Comment 5: Line 547: The authors suggest that since the virus levels are highest during periods of low water temperature, when the metabolism and growth of mussels are reduced, the viruses detected in the mussels are not merely a byproduct of filter feeding. However, could it not be possible that during periods of low water temperature, the degradation and disappearance of viruses in the water are less than during periods of high-water temperature, leading to an increase in virus accumulation in the water and, as a result, an increase in the amount of virus detected in mussels as a byproduct of filter feeding?

Response: This is a great point. We have added additional text to the discussion at lines 555-557 including this possible explanation of the observed results.